# A Method to Design Compact MIMO Patch Antenna Using Self-Isolated Technique

**DOI:** 10.3390/s25072073

**Published:** 2025-03-26

**Authors:** Noi Truong-Quang, Tan Dao-Duc, Phuong Kim-Thi, Tu Le-Tuan, Hung Tran, Dat Nguyen-Tien, Niamat Hussain

**Affiliations:** 1Faculty of Electrical and Electronic Engineering, Phenikaa University, Yen Nghia, Ha Dong, Hanoi 12116, Vietnam; noi.truongquang@phenikaa-uni.edu.vn (N.T.-Q.); tan.daoduc@phenikaa-uni.edu.vn (T.D.-D.); dat.nguyentien@phenikaa-uni.edu.vn (D.N.-T.); 2Faculty of Electrical and Electronic Engineering, Thuyloi University, Hanoi 11515, Vietnam; phuongkt@tlu.edu.vn; 3Department of Convergence Engineering for Intelligent Drone, Sejong University, Seoul 13391, Republic of Korea; huyhung@sejong.ac.kr; 4Department of Intelligent Mechatronics Engineering, Sejong University, Seoul 05006, Republic of Korea; niamathussain@sejong.ac.kr

**Keywords:** MIMO, patch, self-isolated, compact

## Abstract

This paper presents a method to design a compact high-gain multiple-input, multiple-output (MIMO) patch antenna. In a one-dimensional large-scale MIMO array, the conventional approach of using multiple patches and decoupling networks significantly increases the antenna size. To address this, this paper utilizes compact self-decoupled patches, enabling extremely small element spacing while maintaining high isolation levels. Accordingly, a compact size feature can be obtained. Then, gain and bandwidth enhancements are realized with a combination of multiple patches and T-junction dividers. The feasibility of the proposed concept is validated by measurement on a two-element MIMO antenna. The measured results demonstrate that the proposed antennas have good operation characteristics at 4.8 GHz with a small element spacing of 0.008λ. The measured operating bandwidth is about 5%, with isolation of better than 19 dB. A maximum broadside gain of 7.2 dBi can also be yielded.

## 1. Introduction

Multiple-input, multiple-output (MIMO) antennas are among the key components of modern wireless communication systems [1,2]. The critical issue of an MIMO antenna is the mutual coupling between the MIMO elements, which directly deteriorates the matching and radiation performance of the antenna, as well as the communication quality of the whole wireless systems [3]. Among various antenna structures, microstrip patch antennas are widely used due to their benefits of a planar structure, lightweight design, low profile, and ease of integration [4]. However, such antennas typically suffer from low gain radiation. Consequently, mutual coupling suppression and gain enhancement have become research hotspots in the antenna design community.

Various decoupling networks have been proposed for mutual coupling reduction using band-stop filters located between the MIMO elements [5,6,7,8,9]. These band-stop structures block direct coupling from the excited to non-excited element. Another effective solution to improve isolation is establishing additional coupling paths to cancel the original coupling by loading aperture or parasitic elements [10,11,12,13,14]. The third technique is to produce an orthogonal operating mode on the coupled MIMO element by using a near-field resonator [15] or polarization conversion structures [16,17,18,19]. Alternatively, a self-decoupling approach was also investigated in [20,21,22,23,24,25,26]. Although the above-mentioned decoupling structures can achieve satisfactory isolation improvement, large inter-element spacing is required, which significantly increases the overall dimensions when scaling up to multiple-port MIMO systems. Low-gain radiation is another disadvantage of such systems.

For gain enhancement, frequency-selective surface (FSS) layers were proposed in [27,28,29,30,31,32]. These layers are commonly positioned above the radiator, leading to a high profile and non-planar structure. The effective solution is to combine two radiating elements and one T-junction divider as one port of the MIMO system [33,34,35,36]. However, the critical disadvantage of these antennas is that large element spacing is required to achieve high isolation, which makes them less attractive for use in compact devices.

In this paper, a MIMO patch antenna with characteristics of high gain and small element spacing is presented. For the purpose of compactness, the self-decoupled technique is utilized to obtain high isolation with small element spacing. Meanwhile, the gain enhancement is based on the combination of two radiating patches and one T-junction divider as one MIMO port. The proposed concept is validated by measurements on a two-port MIMO antenna. It is also noted that the proposed approach can also work with multiple-port MIMO arrays.

## 2. Contribution of the Proposed Design

The contribution of the presented work is based on the approach to designing an MIMO antenna with features of compact size and high gain. The proposed approach overcomes the limitations of the currently available approaches employed in [27,28,29,30,31,32,33,34,35,36].

The first reported technique employs frequency-selective surface (FSS) layers [27,28,29,30,31,32], which are generally positioned above the radiating patch at proper distances. Nonetheless, this technique considerably increases the antenna dimensions in not only the horizontal direction but also the vertical direction. The second technique utilizes multiple radiating elements, which are excited equally in magnitude and phase with the aid of a T-junction power divider [33,34,35,36]. This method helps to increase the antenna gain while keeping the antenna’s structure planar. The designs proposed in [33,34] do not use any decoupling network for isolation enhancement. Therefore, the MIMO elements are kept at a far distance to achieve high isolation. Meanwhile, the designs proposed in [35,36] employ defected ground structures (DGSs) for mutual coupling suppression. This type of decoupling network limits the ability of the antenna to be integrated into electronic devices, as the DGS is very sensitive to the other adjacent electronic circuits.

This paper utilizes the self-decoupled technique for mutual coupling suppression. The benefit of this technique is significantly reducing the overall antenna’s sizes, which is the result of not requiring extra decoupling networks.. The proposed approach distinguishes itself from previously reported works [27,28,29,30,31,32,33,34,35,36], in which MIMO antennas were generally designed with extremely large overall dimensions. The proposed antenna can also easily be integrated into other electronic circuits, overcoming the limitations of the designs proposed in [35,36]. For gain enhancement, the proposed approach utilizes a method similar to those presented in [33,34,35,36] to achieve low-profile configuration, rather than using FSS layers, as reported in [27,28,29,30,31,32]. The measured data confirm that compared to related works, the proposed approach achieves the smallest MIMO antenna while also achieving reasonable isolation and gain. A further demonstration can be observed in the comparison table in Section 7.

## 3. Compact MIMO Antenna

It is noted that the presented work is a further development of our previous work [26]. However, further gain enhancement and size miniaturization are conducted in this paper. Initially, the design from [26] is selected as the reference MIMO antenna, and its performance is summarized here to provide a clearer understanding of the design process for the compact MIMO antenna proposed in this paper. As discussed, the decoupling mechanism is to produce the fundamental ${\mathrm{TM}}_{01}$ mode on the excited element. Meanwhile, the induced mode on the coupled element is the higher-order mode, ${\mathrm{TM}}_{20}$, which is orthogonal to the mode on the excited element. Accordingly, high isolation can be achieved without requiring any decoupling structure.

Figure 1 shows the configuration of the reference MIMO antenna. Here, two microstrip patches are arranged in the H−plane coupled configuration with an element-to-element spacing of *s* = 1 mm. The fundamental ${\mathrm{TM}}_{01}$ mode (vertical direction) is determined by the patch’s length ($L_{p}$). Meanwhile, the resonance of the ${\mathrm{TM}}_{20}$ mode (horizontal direction) depends on the patch’s width. As the resonance of the high-order mode is generally higher than that of the fundamental mode, a meander-line structure is used to decrease the operating frequency of the ${\mathrm{TM}}_{20}$ mode. Figure 2 shows the performance in terms of the reflection coefficient ($|S_{11}|$) and transmission coefficient ($|S_{21}|$) of the reference design. In general, good operation characteristics are achieved at the desired frequency band around 4.8 GHz.

### 3.1. Patch Size Miniaturization

Apart from the edge-to-edge element spacing, the patch size is also very important, as it determines the compactness of the whole MIMO array. As the resonance of the high-order mode (${\mathrm{TM}}_{20}$) is determined by the width of the patch, three designs with different patch widths of 20, 24, and 28 mm are studied. Figure 3 shows different MIMO designs with different configurations of the patch structures. Theoretically, a smaller patch width results in higher resonance of the ${\mathrm{TM}}_{20}$ mode. Consequentially, more meander-line structures are required to shift this resonance to a lower frequency range.

An S-parameter comparison is provided in Figure 4. It can be observed that these MIMO antennas exhibit good impedance matching and isolation performance at 4.8 GHz. However, a smaller width results in a smaller impedance-matching bandwidth, which is a common feature of microstrip patch antennas. Finally, the case with $w_{p}$ = 20 mm is chosen as the optimal patch for compactness. Note that the patch width can be further decreased; however, this results in a smaller operating bandwidth.

### 3.2. Element-Spacing Reduction

Based on the results achieved in Section 3.1, the most compact MIMO antenna is Design-1 with an edge-to-edge element spacing of *s* = 1 mm, corresponding to about 0.016λ at 4.8 GHz. This spacing is further reduced to demonstrate the usefulness of the proposed self-decoupled technique in designing compact MIMO antennas.

An investigation of the effect of s on antenna performance was carried out with multiple values of s. However, only the smallest value of *s* = 0.5 mm (about 0.08λ) is shown here for brevity. Note that antenna performance is significantly deteriorated when the value of s is smaller than 0.5 mm. Figure 5 shows the final configuration of the proposed MIMO antenna with the smallest patch size and element spacing, designated as Ant-1. Figure 6 shows a performance comparison between Design-1 and Ant-1. It can be obviously seen that both antennas exhibit good operation characteristics around 4.8 GHz.

As a further demonstration of the effectiveness of the presented work, a comparison with the coupled MIMO design with the same element spacing and patch width is shown in Figure 7. The patch length is optimized so that the two antennas have a similar operating frequency of 4.8 GHz. As observed, both antennas show good matching performance at the desired bandwidth. However, the coupled MIMO antenna suffers from serious coupling between the MIMO elements. Meanwhile, the isolation of the self-decoupled MIMO (Ant-1) is 25 dB. The operating bandwidth with both reflection and transmission coefficients of less than −10 dB ranges from 4.77 to 4.83 GHz.

Finally, the current distribution on the presented MIMO antenna at 4.8 GHz is illustrated in Figure 8. The left-side element is excited, while the right-side element is the coupled element. As seen, the operating mode on the excited element is the ${\mathrm{TM}}_{01}$ mode. Meanwhile, that on the non-excited element or coupled element is the ${\mathrm{TM}}_{20}$ mode. This is consistent with the self-decoupling mechanism discussed in [26].

## 4. Compact and High-Gain MIMO Antenna

### 4.1. Design of High-Gain MIMO Element

Since the gain performance of the antenna presented in Section 3 is quite low, at about 5.2 dBi, the approach of using multiple elements and a T divider as an MIMO element is employed. Figure 9 shows the geometry of a high-gain MIMO element, which consists of two compact radiating elements and a T-junction power divider. Here, the divider is printed on another Taconic RF-35 substrate beneath the ground plane.

### 4.2. Design of Two-Port MIMO Antenna

The high-gain MIMO element is used to design a two-port MIMO antenna, designated as Ant-2 in Figure 10. As demonstrated in Section 3.2, the self-decoupled approach can work effectively with an extremely small element spacing of *s* = 0.5 mm; therefore, this value is adopted for the two-port MIMO design with high gain. The design parameters are kept at the optimal values achieved in Section 4.1.

The simulated performance of the proposed high-gain MIMO antenna (Ant-2) is compared with that of the low-gain MIMO antenna (Ant-1 in Figure 5), as shown in Figure 11. As observed, better operating characteristics can be obtained for Ant-2. Ant-1 exhibits an impedance bandwidth of 60 MHz, and the isolation across this band is better than 10 dB. Meanwhile, Ant-2 exhibits a larger impedance bandwidth of 150 MHz, ranging from 4.7 to 4.85 GHz. Within this band, the isolation is always better than 18 dB, and the best isolation of about 40 dB can be achieved at 4.8 GHz. Double resonances observed in the $|S_{21}|$ profile of Ant-2 come from the ${\mathrm{TM}}_{20}$ mode on the unexcited double patches. As the distances from them to the excited patches differ, the resonances are different. In terms of broadside gain, as illustrated in Figure 12, Ant-2 has a maximum gain of 7.4 dBi, which is higher than that of Ant-1 (5.2 dBi).

## 5. MIMO Diversity Performance

To determine the feasibility of deploying the proposed two-port high-gain antenna in MIMO systems, its diversity performance must be thoroughly assessed. This evaluation is conducted based on four key metrics: the Envelope Correlation Coefficient (ECC), Diversity Gain (DG), Total Active Reflection Coefficient (TARC), and Channel Capacity Loss (CCL).

One of the most critical parameters for evaluating MIMO antenna performance is the envelope correlation coefficient (ECC), which quantifies the degree of correlation or isolation between antenna elements. To compute the ECC for a two-port MIMO antenna, a popular approach is to employ S parameters, as shown in Equation (Equation 1). Closely related to ECC, diversity gain (DG) is another essential metric for assessing the reliability and effectiveness of MIMO antennas in wireless communication systems. It measures the improvement in the signal-to-noise ratio (SNR) achieved through diversity techniques, which are vital for enhancing overall system performance. Since the DG is directly dependent on the ECC, it is typically computed using a standard mathematical expression as described in Equation (Equation 2). Lower ECC values signify better isolation and reduced interference, whereas a high DG value indicates minimal correlation between antenna elements, which results in better isolation and improved signal quality. Generally, an ECC value below 0.5 is deemed acceptable, whereas values below 0.01, as illustrated in Figure 13a, indicate superior diversity performance. Additionally, the DG should ideally remain close to 10 dB across the entire operating frequency band, as confirmed by the simulation results in Figure 13b.(1)$$ECC=\frac{{\left|S_{11}^{*}S_{12}+S_{21}^{*}S_{22}\right|}^{2}}{\left(1-{\left|S_{11}\right|}^{2}-{\left|S_{21}\right|}^{2}\right)\left(1-{\left|S_{22}\right|}^{2}-{\left|S_{12}\right|}^{2}\right)}$$(2)$$DG=10\sqrt{1-{\left(ECC\right)}^{2}}$$

In addition to the ECC and DG, the total active reflection coefficient is a crucial parameter for evaluating the efficiency of MIMO antenna systems. It quantifies the amount of power reflected versus the power incident on the antenna ports, with values ranging from 0 to 1, and helps determine how effectively the antenna radiates the input signal while minimizing unwanted reflections. The TARC can be calculated using S parameters and phase angles, as shown in Equation (Equation 3). In addition to the TARC, channel capacity loss is also an important metric, as it measures the reduction in channel capacity due to mutual coupling and correlation between antenna elements. CCL is mathematically derived from the determinant of the correlation matrix and is expressed in bits per second per Hertz (bps/Hz), as expressed in Equation (Equation 4).(3)$$TARC=\sqrt{\frac{{\left|S_{11}+S_{12}e^{j\theta }\right|}^{2}+{\left|S_{21}+S_{22}e^{j\theta }\right|}^{2}}{2}}$$(4)$$CCL=-\log_{2}^{\left|\psi^{R}\right|}$$
where $\left|\psi^{R}\right|=\left[\rho_{ij}\right],\left(i,j\right)\left(1,2\right)$;
and $\begin{array}{c} \rho_{11}=\left(1-{\left|S_{11}\right|}^{2}-{\left|S_{12}\right|}^{2}\right)\\ \rho_{22}=\left(1-{\left|S_{21}\right|}^{2}-{\left|S_{22}\right|}^{2}\right)\\ \rho_{12}=\left(S_{11}^{*}S_{12}-S_{21}^{*}S_{22}\right)\\ \rho_{21}=\left(S_{22}^{*}S_{21}+S_{12}^{*}S_{11}\right) \end{array}$


To achieve optimal performance, TARC should be as close to 0 as possible, ensuring minimal power reflection and maximum transmission efficiency. The simulation result of the proposed MIMO antenna in Figure 14a indicates consistently low TARC values across the bandwidth, highlighting its effectiveness in minimizing unwanted reflections and ensuring reliable communication. Furthermore, a low CCL value signifies better data transmission efficiency, with an optimal threshold typically set below 0.4 bps/Hz. Observation of the calculated result in Figure 14b reveals that the proposed MIMO antenna maintains a CCL much lower than 0.4 bps/Hz across the target frequency band, indicating high-quality signal transmission with minimal disruptions.

## 6. Measurement Results

The design concept is verified by measurements on a fabricated antenna prototype. Photos of the fabricated design showing top and bottom views are shown in Figure 15. For the S parameter, the N5242A vector network analyzer is used to characterize the reflection and transmission coefficients. For far-field features, the antenna is tested in an anechoic chamber. The measured results are generally well matched with the simulated results. However, there always exists a small difference due to the tolerance in fabrication, the antenna assembly, and imperfection in measurement setup.

The simulated and measured reflection coefficients ($|S_{11}|$ and $|S_{22}|$) and transmission coefficient ($|S_{21}|$) are presented in Figure 16. The overlapped −10 dB impedance bandwidth for both Port-1 and Port-2 excitations is from 4.69 to 4.93 GHz, equivalent to about 5%. Meanwhile, the isolation within is always better than 19 dB, with the highest isolation of about 35 dB.

With respect to the far-field results, only measured data with Port-1 excitation are chosen to present due to the symmetrical geometry of the proposed design, leading to similar behavior for both port excitations. Figure 17 shows the simulated and measured broadside gain of the proposed antenna. The measured broadside gain within the impedance bandwidth ranges from 5.4 to 7.2 dBi. The radiation patterns in two principal planes at 4.8 GHz are plotted in Figure 18. In general, the patterns are quite symmetric around the forward direction (θ = $0^{\circ }$). The polarization isolation, which is defined by the gain difference of the co-polarization and cross-polarization, is about 24 dB. Meanwhile, the front-to-back ratio defined by the gain levels in the forward and backward directions is about 15 dB.

Figure 19 shows the calculated ECC based on the S parameter. It can be obviously seen that there is a good match between the simulation and measurement. The measured ECC values in the frequency band of interest are much lower than the accepted value of 0.5.

## 7. Performance Comparison

The advantages of the proposed design can be further demonstrated by making a comparison between the proposed antenna and other related MIMO patch antennas with high-gain radiation, as reported in Table 1. It can obviously be seen that the proposed MIMO antenna has the lowest profile and the smallest spacing between the MIMO elements. This is due to the use of the self-decoupled technique and a compact patch, which does not require additional decoupling networks. Although high-gain radiation was previously achieved in [27,28,30,32], significantly high profiles were caused by the FSS layers. Similar to the method of using two radiating elements for one MIMO port, the designs proposed in [34,36] exhibit higher gain, with a trade-off sacrificing antenna size and spacing. It is worth noting that all reported designs require extremely large element spacings in comparison with the proposed work, which is a problem when scaling the MIMO array to multiple ports. From this perspective, the proposed approach is more beneficial.

## 8. Multiple-Port MIMO Antenna

The ability to scale up to a multiple-port MIMO antenna can be evaluated based on the operation characteristics of a three-port MIMO array, as shown in Figure 20. Figure 21 depicts the simulated S parameter and broadside gain of a three-port MIMO with Port-1 and Port-2 excitations. The data indicate that the antenna can achieve good impedance matching, isolation, and gain in the frequency band of interest. Identical reflection coefficients for Port-1 and Port-2 with a −10 dB bandwidth from 4.7 to 8.5 GHz, which is the same as the two-port MIMO antenna (Ant-2). Meanwhile, the isolations among the MIMO elements are better than 18 dBi, and the broadside gain is higher than 6.7 dBi at 4.8 GHz. Note that Figure 21b shows the realized gain in the broadside direction. The radiation patterns for Port-1 and Port-2 are slightly different due to their different positions.

## 9. Conclusions

A MIMO patch antenna with compact size, high gain, and high isolation has been presented and investigated in this paper. The proposed antenna has very small element spacing, with an edge-to-edge spacing of 0.008λ and center-to-center spacing of 0.33λ. For verification, an antenna prototype was fabricated and measured. All necessary parameters, such as the scattering parameter and far-field characteristics were implemented using vector network analyzer and anechoic chamber. In comparison with other related works, the presented design has a more compact size while maintaining high isolation. The proposed approach also offers advantages for integration with large-scale MIMO arrays.

## Figures and Tables

**Figure 1 sensors-25-02073-f001:**
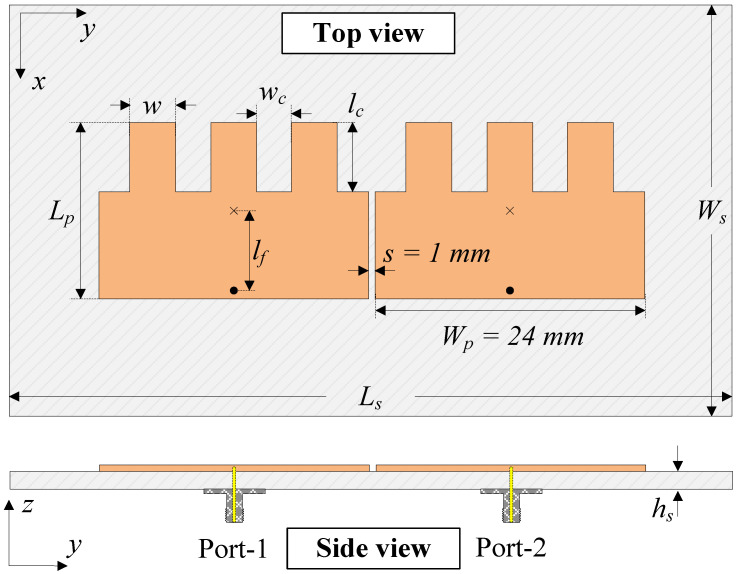
Geometry of the reference MIMO antenna [26]. The design parameters are $L_{s}$ = 70, $W_{s}$ = 40, $h_{s}$ = 1.5, $L_{p}$ = 15.8, $W_{p}$ = 24, $l_{f}$ = 7.5, *s* = 1.0, $l_{c}$ = 6.2, $w_{c}$ = 3.0, and *w* = 4.2 (unit: mm).

**Figure 2 sensors-25-02073-f002:**
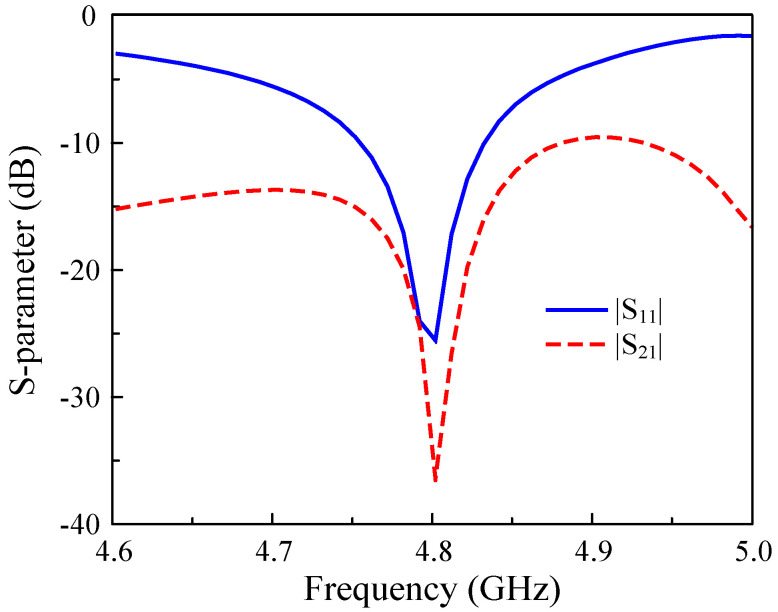
Simulated S parameter of the H−plane coupled MIMO antennas.

**Figure 3 sensors-25-02073-f003:**
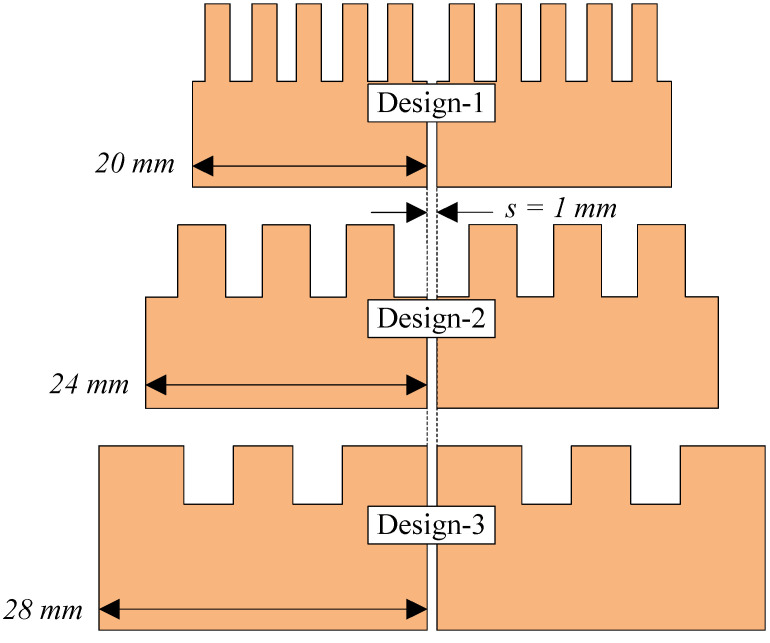
Different MIMO designs with different configurations of the patch structures.

**Figure 4 sensors-25-02073-f004:**
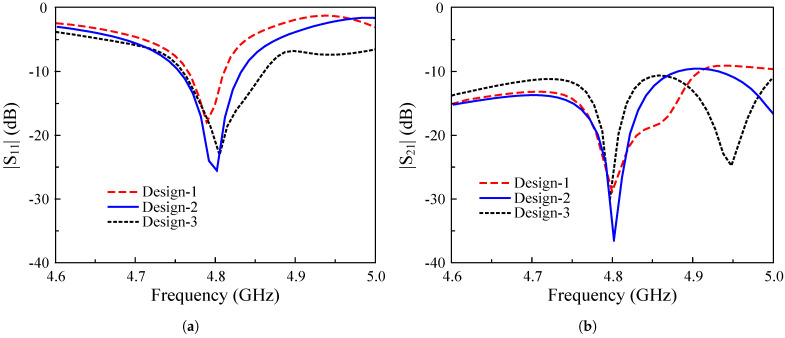
Simulated results of different MIMO antennas. (**a**) $|S_{11}|$; (**b**) $|S_{21}|$.

**Figure 5 sensors-25-02073-f005:**
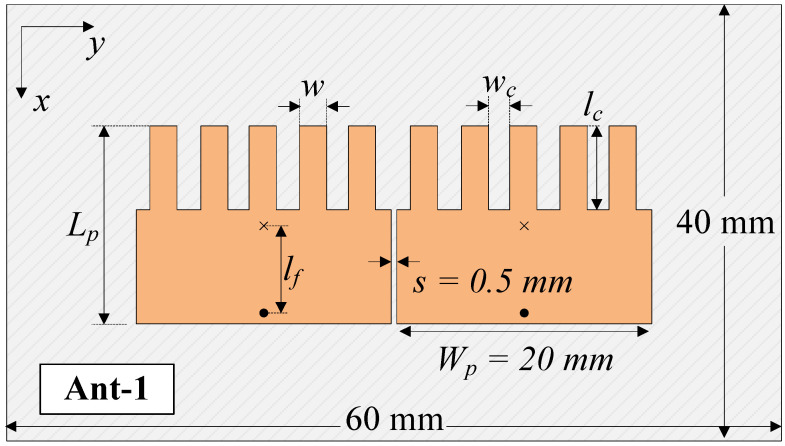
Geometry of the proposed 2-port MIMO antenna with compact size and small element spacing. $h_{s}$ = 1.5, $L_{p}$ = 15.5, $W_{p}$ = 20, $l_{f}$ = 4.9, *s* = 0.5, $l_{c}$ = 6.7, $w_{c}$ = 1.5, and *w* = 2.3 (unit: mm).

**Figure 6 sensors-25-02073-f006:**
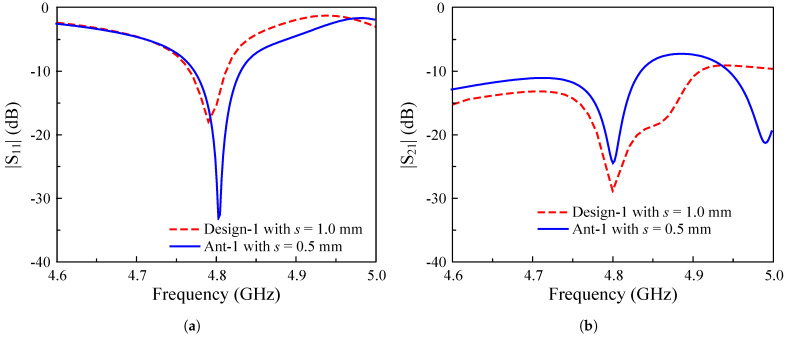
Simulated results of MIMO antennas with different values of s. (**a**) $|S_{11}|$; (**b**) $|S_{21}|$.

**Figure 7 sensors-25-02073-f007:**
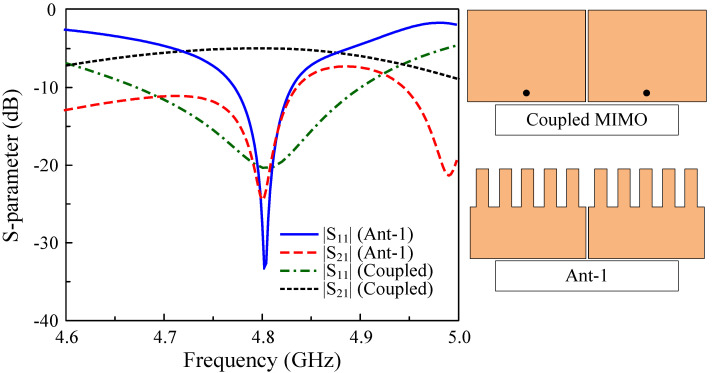
Simulated S parameter of the coupled and self−decoupled MIMO antennas.

**Figure 8 sensors-25-02073-f008:**
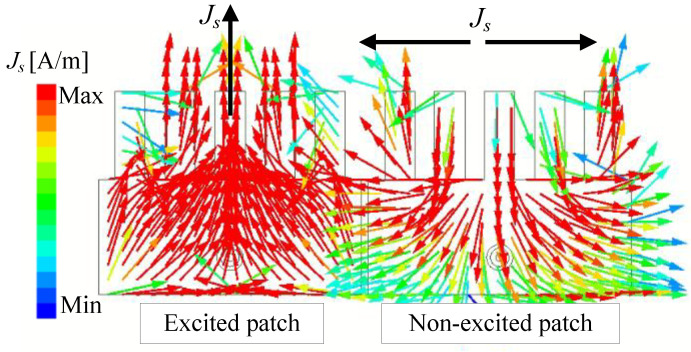
Simulated current distribution at 4.8 GHz.

**Figure 9 sensors-25-02073-f009:**
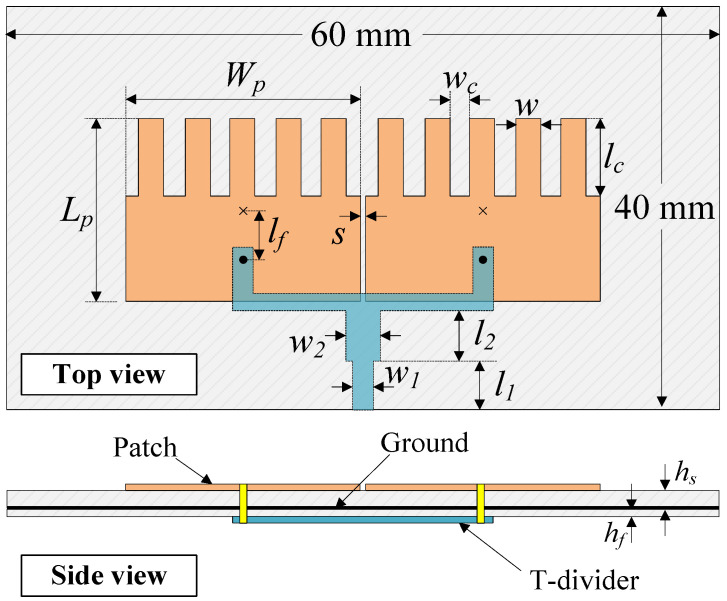
Geometry of the proposed high-gain MIMO element. $h_{s}$ = 1.5, $h_{f}$ = 0.76, $L_{p}$ = 15.5, $W_{p}$ = 20, $l_{f}$ = 4.9, *s* = 0.5, $l_{c}$ = 6.7, $w_{c}$ = 1.5, *w* = 2.3, $w_{1}$ = 1.7, $w_{2}$ = 2.9, $l_{1}$ = 5.0, and $l_{2}$ = 5.2 (unit: mm).

**Figure 10 sensors-25-02073-f010:**
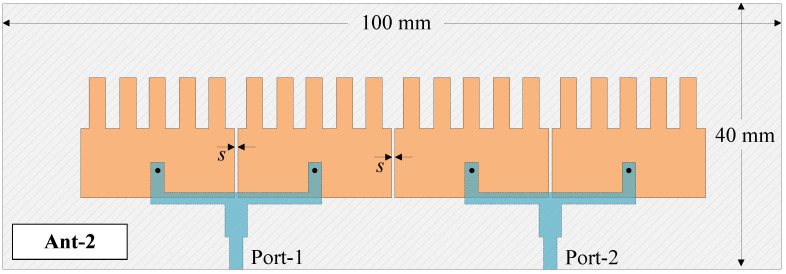
Geometry of the proposed 2-port high-gain MIMO element.

**Figure 11 sensors-25-02073-f011:**
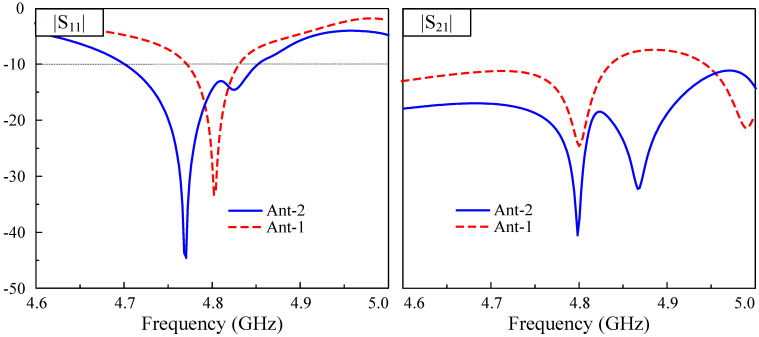
Simulated S parameter of the proposed 2-port high-gain MIMO antenna.

**Figure 12 sensors-25-02073-f012:**
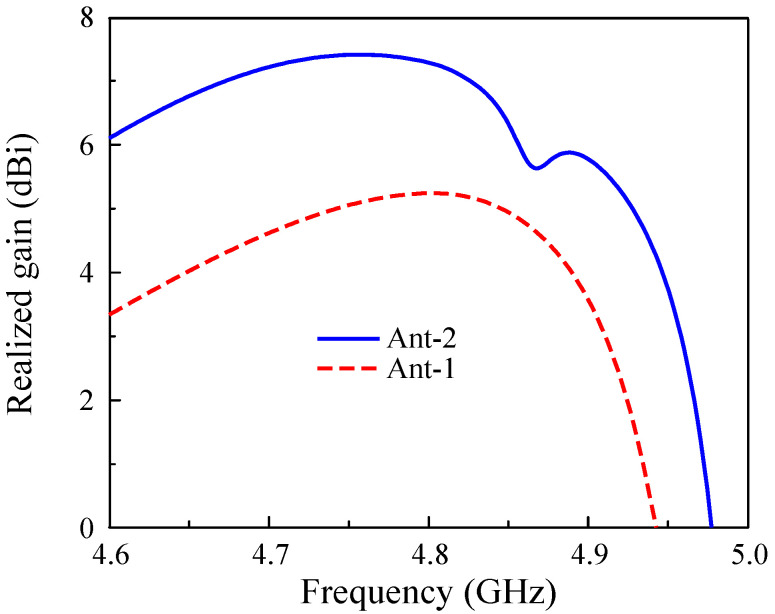
Simulated realized broadside gain of the proposed 2-port high-gain MIMO antenna.

**Figure 13 sensors-25-02073-f013:**
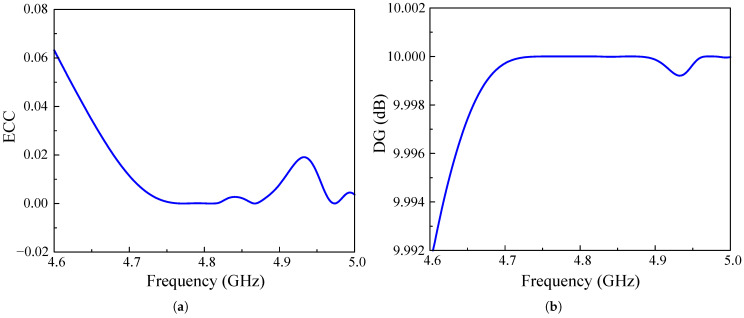
Simulated (**a**) ECC and (**b**) DG of the proposed 2-port high-gain MIMO antenna.

**Figure 14 sensors-25-02073-f014:**
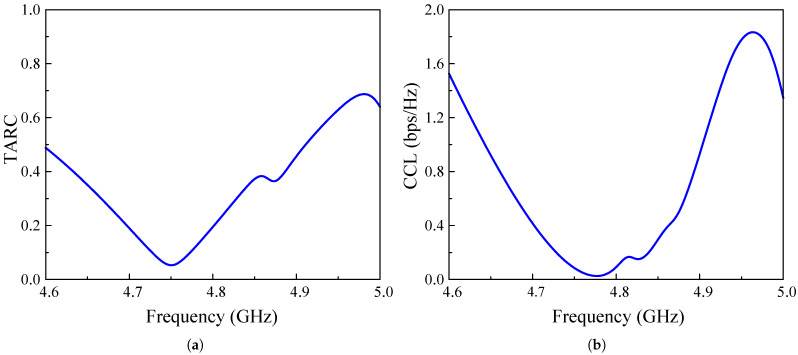
Simulated (**a**) ECC and (**b**) DG of the proposed 2-port high-gain MIMO antenna.

**Figure 15 sensors-25-02073-f015:**
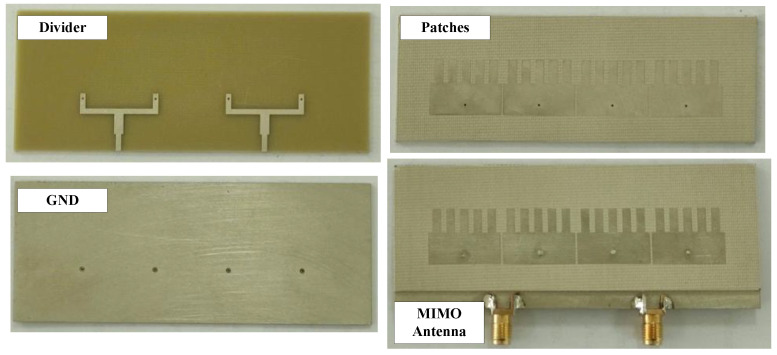
Photographs of the fabricated antenna.

**Figure 16 sensors-25-02073-f016:**
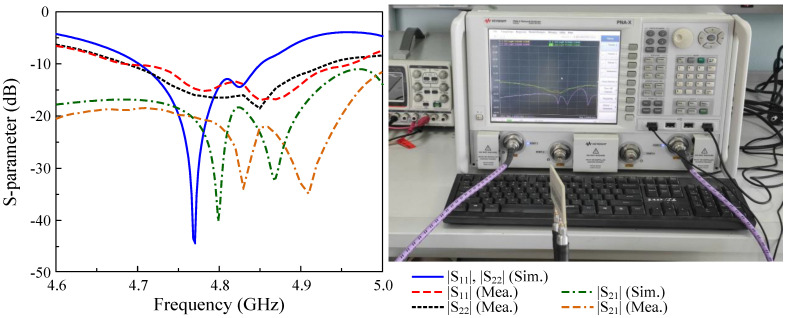
Simulated and measured the S parameter of the proposed antenna.

**Figure 17 sensors-25-02073-f017:**
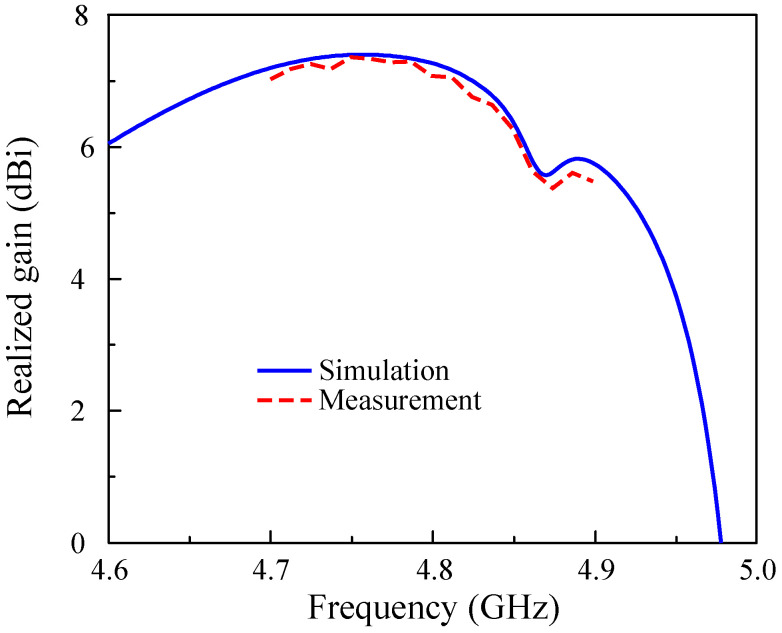
Simulated and measured realized broadside gain of the proposed antenna.

**Figure 18 sensors-25-02073-f018:**
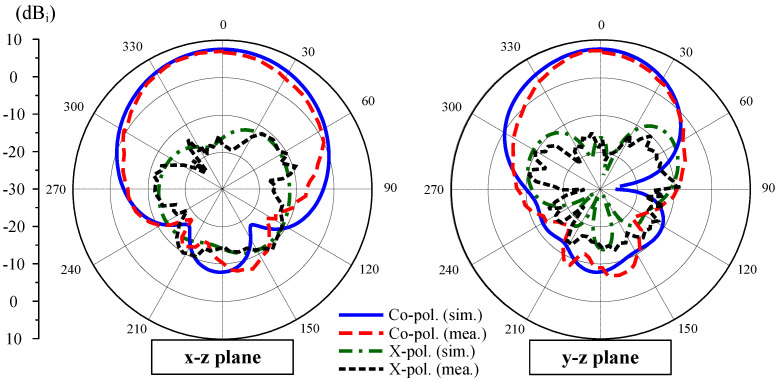
Simulated and measured radiation patterns of the proposed MIMO antenna at 4.8 GHz.

**Figure 19 sensors-25-02073-f019:**
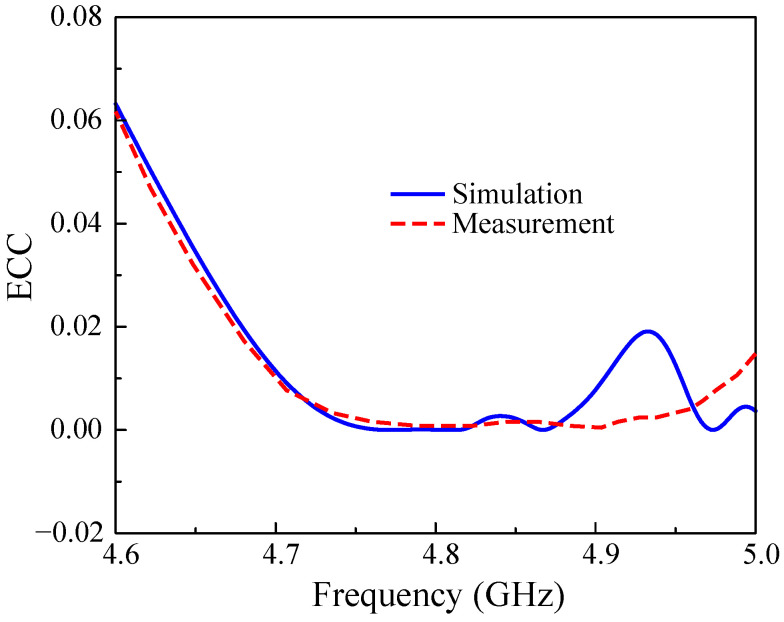
Simulated and measured ECC of the proposed antenna.

**Figure 20 sensors-25-02073-f020:**
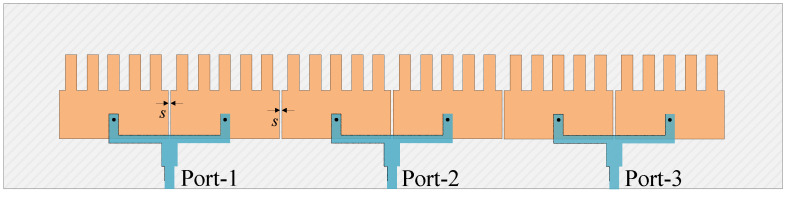
Geometry of the 3-port high-gain MIMO element.

**Figure 21 sensors-25-02073-f021:**
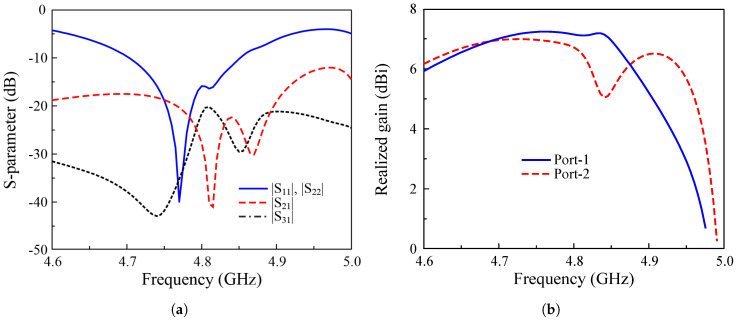
Simulated (**a**) S parameter and (**b**) gain of the 3-port high-gain MIMO element.

**Table 1 sensors-25-02073-t001:** Performance comparison of high-gain MIMO patch antennas.

Ref.	Overall Size (λ)	Decoupling Network	No. of Elements for 1 Port	Center Spacing (λ)	Edge Spacing (λ)	Gain (dBi)	Isolation (dB)
[27]	1.40 × 2.33 × 0.10	No	4	1.11	0.42	11.5	30
[28]	2.60 × 3.73 × 0.64	Yes	2	1.3	0.53	10.3	35
[30]	1.06 × 1.12 × 0.16	No	4	1.19	0.32	14.1	40
[32]	1.84 × 1.84 × 0.27	No	1	0.53	0.13	8.8	10
[34]	2.80 × 3.27 × 0.07	Yes	2	1.41	0.39	12	35
[36]	2.24 × 2.24 × 0.07	No	2	1.22	0.49	10.3	26
Prop.	0.64 × 1.60 × 0.02	No	2	0.33	0.01	7.1	18

## Data Availability

Data are contained within the article.

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
