# Peer review of "A Method to Design Compact MIMO Patch Antenna Using Self-Isolated Technique"

_sensors, 2025, doi:10.3390/s25072073_

Round 1
Reviewer 1 Report
Comments and Suggestions for Authors
This paper presents a decoupling design by using TM01 and TM20 modes. Some detailed comments:
1. The gain enhancement of the proposed antenna is obtained from the construction of a two-element antenna array, which is a conventional approach. Hence, the term "high-gain" in the title may be inappropriate.
2. The proposed work only deals with coupling in the H-plane arrayed antenna rather than the large-scale MOMO array mentioned in the abstract.
3. The operating mode on the excited element is the TM01 mode, while that on the coupled element is the TM20 mode. Why the TM20 mode is not generated on the excited element?
4. In Figure. 20, S11 and S22 are provided with the same curve, while the realized gain under port 1 and port 2 is different. It is unreasonable that the reflection coefficients when fed through port 1 and port 2 are completely same.
5. Please provide the measured ECCs.
In the first paragraph of the Introduction, "due to" can be revised as "because of" or "owing to".
Reviewer 2 Report
Comments and Suggestions for Authors
Review Opinion
sensors-3515019: A method to design compact high-gain MIMO patch antenna using self-isolated thechnique
Authors presented the design of a two-element combined version of their previous work on the MIMO patch antenna. The paper's results are the ones that can easily be extended because
1) It is expected that the isolation in a (1)x(1) configuration of Figure 5 is applicable to a (1x2)x(1x2) configuration of Figure 10 since the coupling geometry is the same.
2) The reduction of the patch width from original 24 mm to new 20 mm is also easily expected since the patch width can be adjusted for sub-square, square, super-square shapes (mode number is zero in the width direction) by suitably positioning the feed probe for impedance matching.
3) The making of a 1x2 configuration with a power divider is straightforward.
I would like to suggest the following improvement.
1. Please consider strengthening the operating principles of the original structure of Figure 1 by adding your analysis on the following issues.
1) Figure 8 shows that the mode in the unexcited patch appears to be a combination of predominantly the TM01 and TM20 modes. The TM20 mode alone could not establish the current distribution on the unexcited patch shown in Figure 8.
2) Then a question arises: Why is the isolation good even with an excitation of the TM01 mode? This question might be answered by investigating the electric field distribution beneath the unexcited patch. The electric field component in the direction of the probe axis should be zero at the probe position for good isolation.
3) There is a possibility that the isolation between two uncorrugated patches in Figure 7 can be made large by a suitable choice of the patch width although the resultant width is larger than that of the corrugated patch.
2. '3.2 Element-spacing reduction'
The width of the prosed antenna is 41 mm = 20 mm*2 + 1 mm with a spacing of 1.0 mm (before reduction). The reduced spacing is 0.5 mm giving the antenna width of 40.5 mm. Therefore, the antenna size reduction is infinitesimal. The point here needs to be 'the sensitivity of isolation versus the spacing'. Please consider shifting the view point.
3. '4. Compact and high gain MIMO antenna'
1) The patch width is reduced from 24 mm to 20 mm, the task of which is simple as I pointed out in the above. Please consider going the limit of the patch width reduction so that your design is really a compact one.
2) I expect your antenna will not be used in free space without any obstructing structures. When installed on a static or portable device, a close proximity of material such as the device case and a human hand might catastrophically destroy the isolation or change the frequency of good isolation. Please consider adding results of investigation on this issue. This can easily be done by simulation.
4. Figure 11:
In Figure 11, there are two resonances in the isolation (S21). Could you explain why double resonances are obtained. Is the TM20 mode excited at two frequencies in the unexcited double patches?
5. Some clarifications are required.
1) Lines 82−83: 'Meanwhile, the operating mode on the coupled will be ...'
the operating mode => the induced mode
2) x mark in Figure 1: It is not clear where is the x mark. Is it at the horizontal and vertical centers of the patch?
3) In Figure 7, the size of the uncorrugated patch appears to be the same as that of the corrugated one. Yet the resonance frequency of both patches is the same. Does the corrugation not shift the resonance frequency of the patch?

Round 2
Reviewer 1 Report
Comments and Suggestions for Authors
All technical concerns have been addressed. No further comments.